# A Scoping Review of Food Literacy Interventions

**DOI:** 10.3390/nu16183171

**Published:** 2024-09-19

**Authors:** Keely O’Brien, Lesley MacDonald-Wicks, Susan E. Heaney

**Affiliations:** 1School of Health Sciences, University of Newcastle, Callaghan, NSW 2308, Australia; keely.obrien@uon.edu.au (K.O.); lesley.wicks@newcastle.edu.au (L.M.-W.); 2Hunter Medical Research Institute Food and Nutrition Research Program, School of Health Science, University of Newcastle, Callaghan, NSW 2308, Australia; 3Department of Rural Health, The University of Newcastle, 20 Highfields Circuit, Port Macquarie, NSW 2444, Australia

**Keywords:** food literacy, interventions, dietary outcomes, programmes, nutrition education, public health, food skills, food, nutrition and diet

## Abstract

Introduction: Food literacy (FL) is a rapidly emerging area of research that provides a framework to explain the interplay of food-related skills, beliefs, knowledge and practises that contribute to nutritional health and wellbeing. This review is the first to scope the current literature for FL interventions, assess their characteristics against the components provided in the most widely cited definition of FL. and describe their characteristics to identify gaps in the literature. Methods: This review scopes original articles describing FL interventions in the Medline, CINAHL, ProQuest Education, Web of Science and AMED databases up to August 2023. Results: Despite the heterogeneity between all seven included studies, they all demonstrated some improvements in their FL outcome measures alongside dietary intake (DI), with the greatest improvements seen in studies that employed a FL theoretical framework in intervention design. Populations at high risk of food insecurity, such as university students and people living in disadvantaged areas, were the main targets of FL interventions. Conclusion: The minimal inclusion of FL theory amongst interventions led to an overall poor coverage of essential FL components, indicating researchers should aim to design future FL interventions with a FL theoretical framework.

## 1. Introduction

Poor-quality dietary patterns and the rising number of diet-related diseases are prominent issues facing the health sector, with an estimated eleven million deaths worldwide attributable to dietary risk factors [1]. Interventions centred around optimising food and nutrition habits are a crucial strategy in addressing the increased burden of non-communicable diseases such as cardiovascular disease and diabetes [2,3]. However, the creation of effective dietary interventions is challenging for researchers, as there is a multitude of complex factors that influence an individual’s dietary habits.

Food literacy (FL) is one approach that provides a comprehensive framework to explain the complex interplay of food-related skills, beliefs, knowledge and practises that contribute to achieving health and wellbeing [4,5,6,7,8]. FL is a burgeoning area of research that currently lacks a universally accepted definition [9]; however, numerous definitions and frameworks have been proposed by various research teams [4,5,6,7,8]. The most popular of these definitions, as identified in a recent scoping review, were those published by Velardo, Cullen et al., Kolasa et al., and Vidgen and Gallegos, who employed various methodologies and perspectives to describe FL [4,5,6,7]. Kolasa et al. contributed one of the earliest conceptualisations of FL by adapting an established definition of health literacy which, in comparison to more modern FL frameworks, provides a one-dimensional explanation of FL that lacks a robust description of FL skills or competencies [7]. More recently, Cullen et al. applied an ecological approach to their results from a scoping review to construct their definition from a community food security and health promotion perspective [6]. Alternatively, Velardo applied Nutbeam’s tripartite model to describe the knowledge and skills encompassed in FL using three levels of literacy: functional, interactive and critical literacy [5]. While these definitions provide valuable perspectives on FL, the most popular amongst these definitions, as identified in several scoping reviews [9,10], is the one crafted by Vidgen and Gallegos, who describe FL as follows [4]:

“The scaffolding that empowers individuals, households, communities or nations to protect diet quality through change and strengthen dietary resilience over time. It is composed of a collection of inter-related knowledge, skills and behaviours required to plan, manage, select, prepare and eat food to meet needs and determine intake” [4] (p. 54).

This definition and associated framework as seen in Table 1, was developed using an iterative process involving a literature review, three-phase Delphi process and semi-structured interviews with disadvantaged young adults to describe four interconnected domains of FL: Plan and Manage, Select, Prepare and Eat [4]. These are further broken down into eleven components that describe the main elements and skills that contribute to these specific domains of FL [4].

This framework is not only the most widely cited [9,10] but is also one of the few definitions that accounts for the broadest range of food and nutrition factors influencing FL as identified in a thematic analysis of FL definitions from a scoping review [11]. A systematic review of FL definitions also highlighted the robustness of Vidgen and Gallegos’s framework by demonstrating its inclusion of all three levels of literacy, as discussed in Velardo’s definition of FL [8]. Therefore, this definition is a clear and comprehensive framework to describe FL and is a suitable benchmark in which to measure the application of the FL construct [4].

While FL is still considered an emerging area of research, it has already provided the foundation for many school-based interventions [12,13,14]. The available literature in this area has been summarised in both systematic and scoping reviews, which have identified FL programmes that create positive outcomes, such as improved food safety knowledge, cooking skills, nutrition knowledge and short-term eating behaviours [12,13,14]. Given the positive effects seen in younger populations, FL interventions have now extended to a wider range of groups, including adults, vulnerable groups and families [15,16,17,18,19,20,21]. A recent scoping review investigated both FL and nutrition literacy (NL) programmes across all age groups to describe their different theoretical frameworks, measures, results and limitations [10]. However, this review excluded people with conditions that affect their eating behaviours, such as type II diabetes, and did not differentiate between FL and NL interventions [10]. Several researchers have proposed that while NL and FL are often used interchangeably, FL encompasses a much broader set of practical and critical skills, whereas NL describes the nutritional knowledge that sits within the components of FL [5,8]. Therefore, exclusively investigating the characteristics of FL interventions and how these relate to the definition provided by Vidgen and Gallegos can provide new insights into the application of FL in a wide range of populations and help guide the future direction of this research [4].

The aim of this review is to map the current evidence relating to FL interventions, assess their characteristics against the components provided in the definition of FL provided by Vidgen and Gallegos and describe their characteristics to identify gaps in the literature [4].

What is currently known in the literature about conducting food literacy interventions?What are the key characteristics of these interventions, including target population, setting and structure?Which components of food literacy are being targeted in these interventions and how do these align with the definition of food literacy described by Vidgen and Gallegos [4]?What are the current gaps in the literature relating to food literacy interventions?

## 2. Materials and Methods

### 2.1. Protocol Registration

This scoping review follows the guidelines published by the Preferred Reporting Items for Systematic reviews and Meta-Analyses extension for Scoping Reviews (PRISMA-ScR) and the framework developed by Arksey and O’Malley [22,23]. The protocol for this review was published on the Open Sciences Framework https://doi.org/10.17605/OSF.IO/GY4TB (accessed on 8 August 2024).

### 2.2. Inclusion and Exclusion Criteria

The inclusion and exclusion criteria for this review were shaped using the Population Exposure Comparison Outcome (PECO) framework.

Studies including participants of all ages and demographics were considered, including studies targeting individuals or groups; however, studies where the population was >50% children (below 18 years of age) were excluded, as these have been the focus of previous reviews. School-based studies and animal studies were also excluded.

Interventions of interest were those that targeted FL or one of the eleven components defined in the definition of FL described by Vidgen and Gallegos [4]. Studies targeting a component of FL as part of a larger intervention were also included if the data for the FL aspect of the intervention could be isolated.

To be included, studies had to have reported a measure of FL, such as a survey or tool. Food, health and nutrition-related outcomes were also included as secondary outcome measures.

Studies published in English and from any date in peer-reviewed journals were eligible for inclusion. Interventional designs such as randomised and non-randomised control trials and pre-post studies were the focus of this review, and observational studies, study protocols and case studies where *n* = 1 were excluded.

### 2.3. Search Strategy and Database Selection

The initial search strategy was developed in consultation with a research librarian by identifying and extracting key words from the research question, objectives, aims and subsequent PECO framework. This initial search strategy was then tested in the Medline database, where it was further refined based on the results of the search. The full search strategy can be found in the Appendix A.

The selection of an appropriate databases and application of the final search strategy for each database was guided by the research librarian to maximise the number of relevant articles located by the search. The search was undertaken in the Medline, CINAHL, ProQuest Education, Web of Science and AMED databases, ending on 29 August 2023.

### 2.4. Study Selection

All results were exported to the EndNote 20 reference management software, where duplicates were removed [24]. The remaining articles were then exported to Covidence screening software [25], where two reviewers (KO, RN) independently screened all article titles and abstracts against the inclusion and exclusion criteria. Conflicts between reviewers were resolved by a third reviewer (SH). The remaining articles underwent full-text screening by two reviewers (KO, SH or RN), with conflicts resolved by discussion with a third reviewer (SH). The reference lists of the included articles were also hand-searched for the identification of relevant articles.

### 2.5. Data Charting

Data were extracted into the data extraction table that was constructed collaboratively with all researchers. Two reviewers (KO, SH) independently extracted and charted the data using a purpose-developed extraction template in Covidence. Data extracted included the following metrics: Authors; Year of Publication; Country; Study Aim/s; Population and sample size; Setting/Context; Methods; Use of theory or framework; Structure of Intervention; Key findings; Limitations; Conclusions. The completed data extraction table can be seen in Table 2. Each included study was also mapped against the four domains and eleven components of FL described by Vidgen and Gallegos based on the content reported in their interventions (see Table 3) [4].

## 3. Results

The five databases yielded 13,499 articles from the search. After 5276 duplicates were removed, 8221 articles remained for title and abstract screening. Once initial screening was completed, 21 studies underwent full-text screening, where 7 articles were included in the final data analysis. This screening process is summarised in Figure 1. During this process, two additional articles that detailed further data analysis of the results from the study by Begley et al. were identified [15,26,27]. For the purpose of this review, results from all three of these studies will be grouped together and reported as one study [15,26,27]. All included studies have been summarised in Table 2. And mapped against Vidgen and Gallegos’s FL framework in Table 3 [4].

### 3.1. Setting and Structure

All studies were published on or after the year of 2020. The most common country in which these studies took place was Australia (n = 3); other countries included Germany (n = 1), Canada (n = 1), Greece (n = 1) and the United States of America (n = 1) [15,19,20]. The structure of the interventions varied widely, with three interventions consisting of weekly, face-to-face groups providing food education and practical activities [15,18,20]. Two interventions involved the use of a FL game; however, one involved a brief, one-off trial of the game and the other asked participants to use the game to plan and select their food-shopping over a three-week period [16,21]. One intervention involved a live-in wellness programme with daily education, meals, dietitian (DT) consults and practical activities [17]. Another intervention involved providing regular education materials and interactive sessions on a social media platform [19].

Most interventions took place over a 3–5-week period (n = 4) [16,17,19,20]. The intervention with the shortest duration time ran for 20 min and the longest took place over 11 weeks [18,21].

### 3.2. Participants

The most common target population for FL interventions was university students (n = 3) [16,18,21], followed by adults, with a specific focus on those living in middle–low-income households and socially disadvantaged areas (n = 2) [15,20]. Cohort sizes varied, with the largest study involving 1092 participants; however, these participants were combined from several intervention cohorts [15]. A majority of the studies (n = 5) featured smaller cohorts ranging from 10 to 39 participants [16,17,18,19,21].

### 3.3. Theoretical Foundation

Three studies used a specific FL model, theory or framework to develop the contents of their intervention [15,17,20], most commonly, the framework and definition published by Vidgen and Gallegos (n = 2) followed by the definition from Krause et al. (n = 1) [4,8,15,17,20]. One study was guided by a nutrition textbook by Sizer et al., but did not explicitly mention a FL theory or framework [16,28].

Other common theoretical frameworks that were used to develop interventions included Social Cognitive Theory (n = 3) [18,20,21], the Health Belief Model (n = 2) [15,16] and Self Determination Theory (n = 2) [20,21].

### 3.4. Food Literacy Measurement Tools and Outcomes

Two studies used validated FL questionnaires to assess the FL of their participants, which included a FL behaviour checklist and the Short Food Literacy Questionnaire (SFLQ) [15,17,29,30]. One study used a shortened version of the FL behaviour checklist [20]. All other studies used a combination of validated tools to create their own FL outcome measure, including two studies that used the General Nutrition Knowledge Questionnaire (GNKQ) combined with either the Health Belief Model Survey (HBMS) or a food safety questionnaire [16,18,19,21,31,32].

Overall, all interventions reported improvements in at least one aspect of their FL scores. Two studies using the FL behaviour checklist and one study using the SFLQ reported significant improvements comparing pre- to post-intervention mean scores for all FL domains (*p ≤* 0.0001, *p ≤* 0.001) and a significant improvement in overall mean scores, respectively (*p* < 0.0001) [15,17,20]. Another study noted improvements in 21–45% of participants on all domains of FL; however, significance was not reported [19]. One study reported improvements in two FL-based behaviours (*p* < 0.05) and four FL-based self-efficacy scores (*p* < 0.05); however, two other FL skills and three self-efficacy factors did not demonstrate improvements [18]. One study reported significant increases in their FL knowledge questionnaire (*p* = 0.002) and another reported significant improvements in their GNKQ scores and two domains of the HBMS (*p* = 0.001, *p* = 0.004, *p* = 0.015); however, when compared to the control, no significant difference was detected for either study [16,21].

Two studies assessed the long-term effects of their interventions on FL scores, with one study applying the FL behaviour checklist three months post-intervention reporting a statistically significant increase in scores for two FL domains (Plan and Manage *p* < 0.0001, Selection *p* < 0.0001) [26]. Another study demonstrated strong, significant improvements in SFLQ scores 18 months post-intervention (β = 0.55, *p* < 0.0001) [17].

### 3.5. Secondary Outcomes

DI was measured in four studies, three of which used self-reported average intakes of fruits and vegetables and the other using the German food frequency questionnaire (GFFQ) [15,17,19,20]. Overall, these measures demonstrated significant improvements, with three studies reporting significant increases in mean vegetable consumption (*p ≤* 0.0001, *p ≤* 0.001, *p* = 0.007) [15,19,20]. However, only two of these three studies identified significant improvements in fruit consumption (*p ≤* 0.0001, *p* = 0.007) [15,19]. The study using the GFFQ reported significant improvements in overall DI [17]. Additionally, one study that tracked the food purchases of participants using their app reported significant improvements in the reduced purchasing of ultra-processed foods when compared to the control group [16]. One study also reported significant improvements in all positive parent feeding practises (*p ≤* 0.001–0.003) using a bespoke questionnaire based on a validated child-feeding questionnaire [20].

### 3.6. Food Literacy Domains and Components

The highest-scoring domain among the interventions was Plan and Manage, which includes the component “Make feasible food decisions which balance food needs with available resources” as the only one reported by all seven interventions [15,16,17,18,19,20,21]. The Eat domain was the second highest-scoring, closely followed by Prepare. The lowest-scoring domain was Select, which also included the lowest-scoring component “Access food through multiple sources and know the advantages and disadvantages of these”.

### 3.7. Plan and Manage

Out of the seven studies, three included all components of this domain [15,18,20]. The component most commonly included as part of an intervention was “Make feasible food decisions which balance food needs with available resources”, which was covered by all studies [15,16,17,18,19,20,21]. “Prioritise money and time for food” was covered by the least number of studies (n = 3) [15,18,20].

### 3.8. Select

Out of the seven studies, only two included all components of this domain [15,19]. The component covered by the most studies (n = 6) was “Judge the quality of food” [15,16,18,19,20,21], with “Access food through multiple sources and know the advantages and disadvantages of these” included in the least number of interventions (n = 2) [15,19].

### 3.9. Prepare

Out of the seven studies, three included both components of this domain [14,17,19], and one study included no components of this domain [15]. The component covered by the most studies (n = 6) was “Make a good-tasting meal from whatever food is available” [14,16,17,18,19,20], followed by “Apply basic principles of safe food hygiene and handling” (n = 3) [14,17,19].

### 3.10. Eat

Out of the seven studies, three included all components of this domain [15,17,19]. Both “Understand food has an impact on personal wellbeing” and “Demonstrate self-awareness of the need to personally balance food intake” were the most commonly included components (n = 5) [15,16,17,19,20,21].

### 3.11. Interventions

The study that included the most FL domains and components was by Begley et al., which covered all four domains and eleven components in their intervention [15]. Most studies included between seven and nine components across three to four domains [17,18,19,20]. The studies with the lowest number of FL components covered in their intervention were Mitsis et al., with four components covered across four domains, and Bomfim et al., with four components across three domains [16,21].

## 4. Discussion

The purpose of this review is to map the current FL evidence and compare the interventions to the FL framework described by Vidgen and Gallegos [4]. This review has identified that both the characteristics and overall inclusion of the different FL domains in these studies are widely varied. The study by Begley et al. was the only one to incorporate all four FL domains and eleven components [15], followed by Tartaglia et al., who included nine components from all four domains [20]. The studies by both Morgan et al. and Ng et al. covered eight different components, closely followed by Meyn et al., who included seven components across all four domains [17,18,19]. Finally, Bomfim et al. and Mitsis et al. scored the lowest with four components; however, Bomfim et al. was the only intervention to not cover an entire domain, Prepare, in their intervention [16,21].

Firstly, this scoping review contributes to the rapidly emerging area of FL. All included studies were published within the 4 years before the search was conducted [15,16,17,18,19,20,21]. This aligns with previous research mapping the use of the term ‘food literacy’ in the academic literature, which identified 83% of all included articles published within the five years preceding their search in 2019 [9].

The lack of fidelity to Vidgen and Gallegos’s framework revealed in the mapping analysis of this review indicates that many FL interventions are likely failing to address essential FL skills, most notably, critical analysis skills [4]. The most common domain addressed in the included interventions was Plan and Manage, which included the most common component “Make feasible food decisions which balance food needs with available resources” [4]. This domain describes a person’s knowledge and skills required to plan and prioritise regular, appropriate food intake within the parameters of their individual needs and resources [4]. This requires the use of fundamental, declarative food and nutrition knowledge and then the procedural application of that knowledge [5]. Conversely, the least common domain featured in the FL interventions was Select, which included the least common component “Access food through multiple sources and know the advantages and disadvantages of these” [4]. This describes a person’s ability to access, use and critically evaluate both their food and food sources [4]. The components in this domain require further, higher-level critical analysis skills, which include the ability to critically appraise food and nutrition information, as well as awareness of the wider sociocultural and environmental impacts of food choices [4,5]. Therefore, interventions that fail to address components within the Select domain are reducing their opportunities to target these crucial, critical analysis skills. Interventions favouring simple nutrition education and skill development are also a trend that has been identified in a recent scoping review of FL and NL programmes, where the most common approaches in their included studies were nutrition education and skill-building activities [10]. This review also revealed very few included studies addressing critical components of NL and FL [10]. This further demonstrates the lack of focus on critical skill building in the current approach of FL interventions. While it is important to foster fundamental food knowledge and skills, future interventions should aim to use FL frameworks such as the one by Vidgen and Gallegos to ensure they are addressing all levels of FL, and its associated skills, comprehensively [4].

This review highlights the limited but growing body of evidence demonstrating how FL interventions produce positive food and nutrition outcomes in populations who are at high risk of food insecurity [33,34,35,36]. The populations most commonly featured in the FL interventions were university students and adults from middle–low-income households and socially disadvantaged areas [15,16,18,20,21]. These populations commonly face food insecurity and other adverse food and nutrition outcomes, with several studies linking university students to poor diet quality and low compliance with dietary guidelines [37,38,39,40]. Similar trends such as high intakes of ultra-processed foods and overall low diet quality are observed among low-income earners and people living in areas with the greatest levels of socioeconomic disadvantage [41,42,43]. Dietary interventions amongst these populations have so far achieved mixed results; one systematic review and meta-analysis examined 35 studies targeting behavioural interventions for low-income earners and found that overall effects on diet were small but positive [44]. However, when examining the relationship between socioeconomic position and healthy eating interventions, a recent review found that downstream approaches such as dietary counselling increased socioeconomic inequalities [45]. Amongst university populations, a systematic review of dietary interventions found that 13 of the successful studies focused on either self-regulation skills and nutrition education or environmental modification, such as point-of-sale messaging; however, data on the long-term outcomes of these interventions were limited [46]. This evidence indicates there is currently no optimal approach when creating well-rounded food and nutrition interventions for populations who are at high risk of food insecurity. However, this review has provided evidence that FL is one holistic approach that considers both food and wider social inequities to achieve positive outcomes in these populations [15,16,17,18,19,20,21]. Given this evidence, further investigation of the benefits of taking a FL approach with other vulnerable groups experiencing complex food-related barriers, such as culturally and linguistically diverse groups or low-income earners, is warranted.

Unsurprisingly, the results of this review suggest that interventions where a FL framework was used as the theoretical foundation for study design produced stronger improvements in their FL outcomes when compared to those that did not use FL theory [15,16,17,18,19,20,21]. The studies by Begley et al. and Tartaglia et al. used the Vidgen and Gallegos FL framework and therefore score highest in the intervention mapping analysis in Table 3 [4,15,20]. The other study that utilised FL theory in their intervention was by Meyn et al., who used the definition by Krause et al., which describes an extensive list of FL components based on a systematic review of other FL definitions [8,17]. These three interventions all reported strong, significant improvements across all FL outcomes [15,17,20]. Furthermore, the studies by Begley et al. and Meyn et al. demonstrated the sustainability of these improvements at follow-ups 3–18 months post-intervention [15,17,20]. It should be noted, however, that none of these three studies included a control group [15,17,20]. Other interventions that lacked the inclusion of FL theory, such as the studies published by Mitsis et al. and Bomfim et al., scored the lowest on the mapping analysis and reported mixed results [16,21]. Both of these studies did report some improvement in their FL measures; however, they were deemed non-significant when compared to the control [16,21]. The study by Ng et al. also lacked a FL theoretical foundation but reported improvements across all FL domains; however, they failed to report the statistical significance of their outcomes [19]. The intervention by Morgan et al., who again did not include a FL framework in the development of their intervention, achieved significant improvements across some, but not all, of their FL outcome measures [18]. Overall, these results demonstrate a clear advantage in using a FL theoretical framework to construct FL interventions to produce superior outcomes; however, further interventions including control groups are also needed.

This review has many strengths, including it being the first to comprehensively scope the literature for interventions targeting FL using specific FL outcomes in a non-school-aged population. The authors complied to both the scoping review guidelines by Arksey and O’Malley and the PRISMA extension for scoping review guidelines, which supports the validity of the search and reporting of the results [22,23]. However, despite following rigorous methodology, the following limitations should be considered when interpreting the results of this review. As FL is an emerging area of research, this review only included a small number of studies. Furthermore, many of the key characteristics such as the setting, structure, theoretical basis and measurement tools used to assess FL varied between interventions. These differences, particularly differences in the theoretical basis of FL used to score participants, limits the comparability of outcomes across studies. Additionally, some studies also lacked detailed reporting of their interventions, which led to difficulties when interpreting and mapping the intervention components.

## 5. Conclusions

This review identified some common trends within FL interventions such as small cohort sizes, three- to five-week durations and a focus on populations who are at high risk of food insecurity; however, the study characteristics were largely heterogenous, which limited the comparability of outcome measures between studies. Despite this, all studies achieved some improvement in their FL outcome measures with DI, particularly vegetable intake, also seeing significant improvements across most studies [15,16,17,18,19,20,21]. Furthermore, the few studies that reported follow-up measures demonstrated long-term improvements for most FL and some dietary outcome measures [17,26]. The mapping analysis revealed that most interventions failed to comprehensively cover all FL domains and components included in the Vidgen and Gallegos framework, meaning crucial FL skills, such as critical analysis skills, were not being adequately addressed [4]. Interventions that were designed using a FL theoretical framework or definition were in the minority; however, those that did appeared to yield stronger FL outcomes [15,17,20]. This demonstrates the need for future FL interventions to be designed using a robust FL framework, such as the Vidgen and Gallegos framework, to ensure they encompass the wide range of knowledge and practical skills needed to comprehensively promote FL. These interventions should include validated FL measurement tools to ensure that all aspects of FL are being captured and assessed. Further studies should also aim to work with disadvantaged populations at high risk of food insecurity, have larger cohort sizes, include long-term follow-up measures and include control groups to confirm and further explore the benefits of this area of research.

## Figures and Tables

**Figure 1 nutrients-16-03171-f001:**
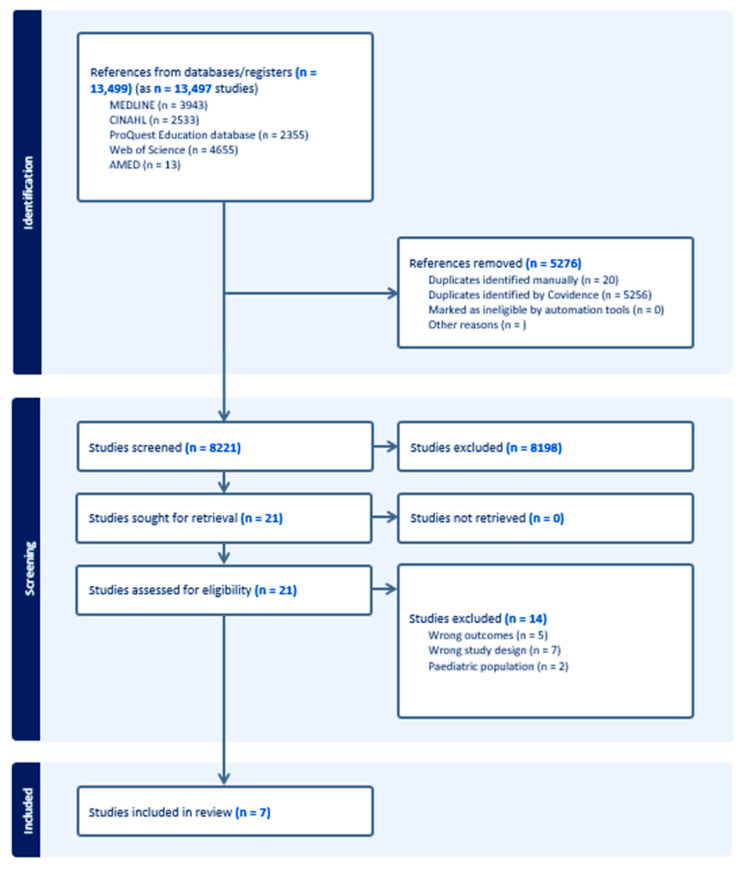
PRISMA flowchart of search.

**Table 1 nutrients-16-03171-t001:** The four domains and eleven components of food literacy (FL), defined by Vidgen and Gallegos [4] (p. 55).

1. Plan and Manage	2. Select	3. Prepare	4. Eat
1.1 Prioritise money and time for food. 1.2 Plan food intake (formally and informally) so that food can be regularly accessed through some source, irrespective of changes in circumstance or environment. 1.3 Make feasible food decisions which balance food needs (e.g., nutrition, taste, hunger) with available resources (e.g., time, money, skills, equipment).	2.1 Access food through multiple sources and know the advantages and disadvantages of these. 2.2 Determine what is in a food product, where it came from, how to store it and how to use it. 2.3 Judge the quality of food.	3.1 Make a good-tasting meal from whatever food is available. This includes being able to prepare commonly available foods, efficiently using common pieces of kitchen equipment and having a sufficient repertoire of skills to adapts recipes (written or unwritten) to experiment with food and ingredients. 3.2 Apply basic principles of safe food hygiene and handling.	4.1 Understand that food has an impact on personal wellbeing. 4.2 Demonstrate self-awareness of the need to personally balance food intake. This includes knowing foods to include for good health, foods to restrict for good health, and appropriate portion size frequency. 4.3 Join in and eat in a social way.

**Table 2 nutrients-16-03171-t002:** Summary and key characteristics of each included study.

Lead Author and Year Title Country	Study Design Aim	Participants Sample Size Setting	Theoretical Basis	Structure Duration	Food Literacy Outcome Measure	Secondary Outcome Measures	Results: Primary Outcome	Results: Secondary Outcome	Study Limitations
Begley 2019 Effectiveness of an Adult Food Literacy Program [15] Australia	Pre-post study Assess how effective the Food Sensation for Adults programme is in changing FL and selected dietary behaviours.	Adults from low–middle-income households who would like to increase their FL skills n = 1092 Community-based groups and virtual sessions for regional areas	Vidgen and Gallegos FL model (Vidgen and Gallegos, 2014), Best Practice Criteria for Food Literacy Programs (WA Department of Health), Health Belief Model, Social Learning Theory	I = 2.5 hr sessions. Four core modules (healthy eating, food safety, cooking, label reading, food selection, budgeting, and meal planning) taught over 3 sessions. Recipe book provided to all participants. Four sessions	Validated pre/post-programme questionnaires: 14-item FL behaviour checklist.	Four close-ended questions on dietary behaviours: average consumption of fruit and vegetable servings, frequency of fast food meals and sugar-sweetened drink consumption (self-reported)	Significant ↑ in all three assessed FL behaviour factors (all *p* ≤ 0.0001) from pre- to post-programme.	Significant ↑ in servings of fruits (*p* ≤ 0.0001) and vegetables (*p* ≤ 0.0001), comparing pre- to post-programme. Significant decreases in fast food meal consumption pre- to post-programme.	Self-selection bias, number of questions assessing domains of FL were limited, potential that culturally and linguistically diverse populations not represented in evaluations, self-reported data, no control
Begley 2020 Identifying who improves or maintains their food literacy behaviours after completing an adult program [26]	Cross-sectional Compare demographic characteristics of participants who completed the programme’s follow-up questionnaire three months after programme completion and assess whether FL and dietary behaviour changes were improved or maintained	n = 621					Mean scores for 2 of the 3 domains significantly ↑ (Plan and Manage, *p* < 0.0001. Selection, <0.0001) from end-of-programme to follow-up. Preparation scores decreased but remained significantly ↑ from baseline.	Servings of fruit and veg decreased but remained significantly ↑ from baseline. Intake of fast food meals significantly ↑ between end-of-programme and follow-up (*p* < 0.0001), consumption frequency decreased from beginning of programme (*p* < 0.0001). No change in frequency of sugar-sweetened drinks.	Unknown what the ideal time for follow-up is; the demographic characteristics of those completing the follow-up questionnaires are different.
Dumont, C 2021 Effectiveness of Foodbank Western Australia’s Food Sensations ^®^ for Adults food literacy program in regional Australia [27]	Cross-sectional Determine if there are differences in the effectiveness of FSA in regional and metropolitan (metro) participants	n = 1849					Significant ↑ in post-programme scores for all three FL domains for metro and regional (*p* < 0.0001). Regional significantly ↑ in selection behaviours compared to metro (*p* < 0.01). No significant difference between metro and regional in other 2 domains	Significant ↑ (*p* < 0.0001) in fruit and vegetable serving intake for metro and regional. Fast food meal and sweetened beverage intake significantly decreased pre- and post-programme for metro (*p* < 0.0001), but not for regional.	May not have captured the full range of disadvantage in regional areas in Western Australia
Bomfim 2020 Food Literacy while Shopping: Motivating Informed Food Purchasing Behaviour with a Situated Gameful App [16] Canada	Other: Exploratory Field Study To investigate the effectiveness of a gameful-situated app ‘Pirate Bri’s Grocery Adventure’ (PBGA) to promote FL in young adults	University students 18–31 Y n = 24: 2x cohorts of 12 Use of app during shopping trips for groceries	Nutrition: concepts and controversies (4th ed) (Sizer et al.), Meaningful gamification, slow technology, Health Belief Model	I = PBGA app C = My Food Guide app Both groups used app to plan and select foods for 3 weeks on minimum 3 different days 3 weeks	General Nutrition Knowledge Questionnaire (GNKQ) and Health Belief Model Survey (HBMS)	Food Purchases	GNKQ: Average scores ↑ 55.17/88 to 59.38/88 from pre- to post-intervention, *p* = 0.001. No differences in post-intervention scores between I and C (*p*> 0.005). HBMS: ↑ from pre- to post-intervention scores for self-efficacy (*p* = 0.004) and perceived susceptibility (*p* = 0.015). No significant difference between scores for I and C for all sections of HBMS.	↑ in fruit and veg (*p* = 0.004) purchased compared to what was planned, no difference between I and C. ↑ in ultra-processed foods bought compared to planned for C (*p* = 0.13), but not for I.	Does not provide insight into clinical effectiveness to promote FL; budgeting not addressed
Tartaglia 2023 Effectiveness of a food literacy and positive feeding practices program for parents of 0 to 5 years olds in Western Australia [20] Australia	Pre-post study To describe the development and evaluation of an innovative programme that combines FL with positive parent feeding practises, targeting parents indisadvantaged areas of Western Australia	Parents of children 0–5 years in Western Australia, particularly those in socially disadvantaged areas >18 Y n = 224 Community and online group sessions	Vidgen and Gallegos FL model (Vidgen and Gallegos, 2014), Satter eating competence model, division of responsibility framework, self-determination theory framework, social cognitive theory (SCT)	I = Weekly education and cooking sessions on basic nutrition principles for the whole family, child-feeding development stages, strategies to overcome fussy eating, food safety, label reading, meal planning, food shopping and budgeting. Includes 60 min hands-on learning, 60 min cooking and 30 min eating. Face-to-face: 5 weeks Online: 4 weeks	Pre- and post-questionnaire comprised 13 items from a 15-item validated FL tool	Positive parent feeding practises: 10 questions from validated child-feeding questionnaires, including the Feeding Practices and Structure Questionnaire. Typical vegetable intake over previous month	Statistically significant ↑ in all FL behaviours (*p ≤* 0.001)	Significant ↑ in all positive parent feeding practises (*p ≤* 0.001–0.003). Significant mean ↑ in vegetable intake (*p* = 0.001)	Higher rate of females (98%); change to online delivery may have resulted in people from higher socioeconomic areas being recruited.
Morgan 2023 Assessing food security through cooking and food literacy among students enrolled in a basic food science lab at Appalachian State University [18] United States	Pre-post study Implement a FL-based curriculum to increase FL-based skills and self-efficacy and combat food insecurity among undergraduate students enrolled in an already-established Basic Food Science Laboratory course at a rural university located in the Appalachian region	University students n = 39 University course	SCT, experiential learning theory	I = University food science course including labs involving observation and hands-on food preparation, food safety, budgeting education and eating 11 weeks	Purpose-developed questionnaire based on a variety of validated instruments	Food security: modified version of the USDA Six Item Food Security Short Form	Significant ↑ from pre- to post-assessment for FL-based behaviours (*p* < 0.05): preparing and cooking a meal with raw ingredients (*p* = 0.039), proper food storage (*p* = 0.046) and FL-based self-efficacy: Using different cooking methods, (*p* = 0.037), cooking with raw or basic ingredients (*p* = 0.003), preparing a well-balanced meal (*p* = 0.018), using substitutions in recipes (*p* = 0.000) and meal planning (*p* = 0.009). Two FL skills and three self-efficacy factors did not see a significant improvement.	No significant improvement in food security indicators	Short study length, small sample size, generalizability not tested, low number of behaviour-focused questions, self-report measures, significant proportion of included participants were dietitian (DT) and fermentation students
Mitsis 2019 Evaluation of a Serious Game promoting Nutrition and Food Literacy: Experiment Design and Preliminary Results [21] Greece	Quasi-experimental trial Present the experiment design and the obtained preliminary results from the evaluation of Express Cooking Train, a serious game that focuses on promoting nutrition literacy (NL) and FL	University students n = 29: Group A 9, Group B 10, Control 10 Trial of computer game	World Health Organisation and the American Heart Association fact sheets	I = FL game with two stages. User experiments with ingredients and progresses in the game when preparing healthy meals. C = Reading nutrition fact sheets 20 min	Knowledge questionnaire based on the (GNKQ) and a validated food safety knowledge questionnaire.	NA	Significant ↑ in pre- to post-intervention knowledge questionnaire scores (*p* = 0.002). No significant difference between C and I group post-intervention scores.	NA	Small sample size, control game was a different format
Ng 2022 Assessing the effectiveness of a 4-week online intervention on food literacy and fruit and vegetable consumption in Australian adults: The online MedDiet challenge [19] Australia	Pre-post study Develop and trial an online intervention programme to improve FL and fruit and vegetable intake through the use of MedDiet principles	Members of the GMHBA private health insurance provider from Victoria >18 Y n = 29 Facebook group	NA	I = Moderators shared nutrition education and encouraged Mediterranean-style eating via infographics 3x per week, how-to-videos 1x per week and recipes 1x per week. Fortnightly Q&A with nutrition experts on a Facebook group. Participants received a box of MedDiet staple ingredients, recipes and cooking ideas. 28 days	Modified version of validated 11-item FL questionnaire from the EFNEP	Average daily fruit and veg consumption: two questions from the National Nutrition Survey	Percentage of participants ↑ in all 11 FL components ranging from 20.7 to 44.8%: comparing prices (20.7%), changing recipes (24.1%), trying a new recipe (24.1%), confidence with cooking variety (31%), food labels (27.6%), nutrition information panel (34.5%), managing money to buy healthy food (31%), consideration of healthy choices (27.6%), including food groups (44.8%), making shopping lists (27.6%), planning meals (20.7%)	Statistically significant ↑ in mean fruit (*p* = 0.021) and veg intake (*p* = 0.007).	Small sample size compared to power calculation, majority tertiary-educated population, sample recruited from staff/members of private health insurer, self-report bias
Meyn 2022 Food Literacy and Dietary Intake in German Office Workers: A Longitudinal Intervention Study [17] Germany	Longitudinal Intervention Study Investigate the 1.5-year long-term effectiveness of a 3-week full-time workplace health-promotion programme (WHPP) regarding FL and dietary intake (DI), as well as therelation between FL and DI of German office workers using four measurement time points.	Adult office workers n = 144 WHPP based at a hotel	FL definition by Krause et al. 2018, Information–motivation–behaviour skills model	I = Groups provided initial and long-term goal-setting with a DT, provided meals with nutrition information and portion sizing guide with DT present, 4 hr nutrition education workshop and individual sessions by DT, behaviour change and health risk presentations, nutrient tables and recipes to take home 3 weeks	Short Food Literacy Questionnaire (SFLQ) adapted to German. Measured pre- (T0) and post (T1)-intervention, 6 (T2) and 18 months (T3) post-intervention	DI: German Food Frequency Questionnaire (GFFQ)	Strong ↑ in FL at T1 (β = 0.52, *p* < 0.0001), T2 (β = 0.60, *p* < 0.0001) and T3 (β = 0.55, *p* < 0.0001).	DI scores ↑ from 13.7 at T0 to 19.3 at T1, then decreased to 15.4 at T2 and 15.3 at T3. Significant ↑ at T1 (β = 0.63, *p* < 0.0001) and weak ↑ at T2 (β = 0.10, *p* < 0.05) and T3 (β = 0.10, *p* < 0.05)	No control/comparison, self-reported measures, T3 and T4 recorded during COVID-19, primarily highly educated population, narrow focus on FL for their specific study aim

Key: I = intervention, C = control, FL = food literacy, Y = age in years, ↑ = increase, DT = dietitian, SCT = social cognitive theory.

**Table 3 nutrients-16-03171-t003:** Mapping of each FL intervention against the four domains and their respective components described in the FL framework by Vidgen and Gallegos [4].

First Author, Year	Begley, A, 2019 [14]	Bomfim, M, 2020 [15,16]	Meyn, S, 2022 [16]	Mitsis, K, 2019 [20]	Morgan, M, 2023 [17]	Ng, Ashley, 2022 [18]	Tartaglia, J, 2023 [20]	Total Components Included across all Interventions for Each Domain
1. Plan and Manage
1.1 Prioritise money and time for food	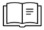	0	0	0	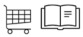	0	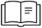	
1.2 Plan food intake	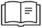	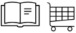	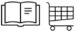	0	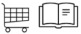	0	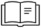	
1.3 Make feasible food decisions which balance food needs with available resources	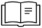	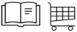	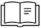	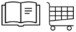	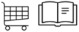	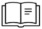	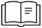	
Total included for Plan and Manage domain	3	2	2	1	3	1	3	15/21
2. Select
2.1 Access food through multiple sources and know the advantages and disadvantages of these	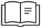	0	0	0	0	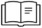	0	
2.2 Determine what is in a food product, where it came from, how to store it and how to use it	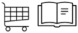	0	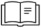	0	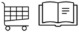	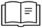	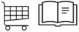	
2.3 Judge the quality of food	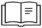	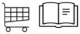	0	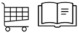	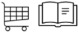	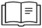	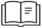	
Total included for Select domain	3	1	1	1	2	3	2	13/21
3. Prepare
3.1 Make a good-tasting meal from whatever food is available.	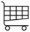	0	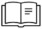	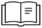	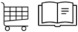	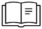	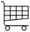	
3.2 Apply basic principles of safe food hygiene and handling	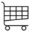	0	0	0	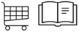	0	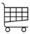	
Total included for Prepare domain	2	0	1	1	2	1	2	9/14
4. Eat
4.1 Understand food has an impact on personal wellbeing	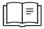	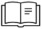	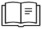	0	0	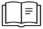	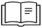	
4.2 Demonstrate self-awareness of the need to personally balance food intake.	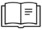	0	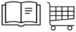	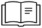	0	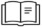	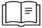	
4.3 Join in and eat in a social way	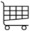	0	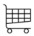	0	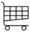	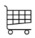	0	
Total included for Eat domain	3	1	3	1	1	3	2	14/21
TOTAL COMPONENTS	11	4	7	4	8	8	9	

Table 3 describes if and how each component and domain of Vidgen and Gallegos’s FL framework was covered in each included intervention [4]. 
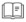
 = Theory and education were included for that component, e.g., informative presentations, written educations materials provided. 
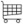
 = Skill development and practical activities for that component were included, e.g., cooking classes, label-reading activities. 0 = Component was not covered as part of the intervention. Numerals refer to the number of components covered by an intervention for each domain and the total number of components covered out of 11. A tally is also provided for the total number of components covered in each domain across all interventions. The tally provides a score out of 21 for the Plan and Manage, Select and Eat domains, and out 14 for the Prepare domain.

## Data Availability

The original contributions presented in the study are included in the article and Appendix A, further inquiries can be directed to the corresponding author.

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
