# Peer review of "A Scoping Review of Food Literacy Interventions"

_nutrients, 2024, doi:10.3390/nu16183171_

Round 1

Reviewer 1 Report

Comments and Suggestions for Authors

Dear Authors,

The manuscript submitted for review (ID: nutrients 3172957) is a very interesting review of the literature with a very well-planned search methodology and an enormous substantive and intellectual contribution of the authors to the preparation of the manuscript and the analysis of literature data available in the databases: MEDLINE, CINAHL, ProQuest Education, Web of Science, AMED.

 From the critical comments:

1. Title: I think it should be changed to a more catchy one, as it may be incomprehensible.

2. The Introduction section seems relatively modest, considering the number of articles included in the analysis.

3. In Table 2, it seems there should be a single space because it is a table on 8 pages, and it is tough to focus on the content of the table, which, in my opinion, is very interesting.

 Table 3 is particularly interesting.

The proposed conclusions are formulated correctly.

References used to prepare the manuscript were selected correctly and properly cited following the Journal's rules.

Reviewer

Author Response

Thank you for considering our manuscript. We appreciate the feedback provided and have addressed the reviewer's comments below and by making amendments to the manuscript (highlighted in yellow in the document). 

Reviewer comment 1 Title: I think it should be changed to a more catchy one, as it may be incomprehensible.

Thank you for your comment, however we believe the current title should not be changed as it clearly communicates the subject matter of the manuscript. While we agree the title is not catchy, it concisely describes the focus and type of study to the reader. This directly aligns with the journal guidelines regarding manuscript preparation “The title of your manuscript should be concise, specific and relevant. It should identify if the study reports (human or animal) trial data, or is a systematic review, meta-analysis or replication study.”  

The Introduction section seems relatively modest, considering the number of articles included in the analysis.

Thank you for your comment. We agree and have therefore provided further background into FL theory in the introduction. These revisions can be found in highlighted text on pages 1 and 2, lines 37-74, where we have provided more detailed background information relating to Food Literacy and a comparison of the most frequently cited definitions.

In Table 2, it seems there should be a single space because it is a table on 8 pages, and it is tough to focus on the content of the table, which, in my opinion, is very interesting

Thank you for your feedback. We agree with this comment, therefore, we have adjusted the line spacing for Table 2. This can be found on pages 7-11. 

Reviewer 2 Report

Comments and Suggestions for Authors

The review on Food Literacy (FL) presented by O’Brien and colleagues provides a valuable overview of the current literature on food literacy.

First, while Vidgen and Gallegos’ definition of FL is widely cited, relying solely on this framework may limit the scope of the review. It would be beneficial to incorporate alternative or complementary definitions and frameworks to provide a more comprehensive analysis of FL interventions.

The review suggests that all studies showed some improvement in FL outcomes, but it doesn't critically evaluate the magnitude of these improvements. The discussion on the extent of the improvements and the potential limitations in measuring FL outcomes would provide a clearer picture of the effectiveness of these interventions.

The focus on populations at high risk of food insecurity is important, but the review could explore whether these interventions are equally effective across different demographic groups. Are there specific subgroups within these populations that benefit more or less from the interventions? 

 The review highlights the use of FL theoretical frameworks in intervention design but doesn’t sufficiently critique the quality or appropriateness of these frameworks.

While the review identifies gaps, such as the minimal inclusion of FL theory in interventions, it should provide more concrete recommendations for future research. What specific aspects of FL theory should be incorporated into future interventions? How can researchers better measure and evaluate FL outcomes?

Also, the review acknowledges the heterogeneity of study characteristics (e.g., cohort sizes, durations, target populations), but it doesn’t explore the implications of this variability on the generalizability or comparability of results.

The review, mainly in conclusions, should also emphasize the need for more rigorous and long-term follow-up studies to confirm these findings. 

Author Response

Thank you for considering our manuscript. We appreciate the feedback provided and have addressed the reviewer's comments below and by making amendments to the manuscript (highlighted in yellow in the document). 

First, while Vidgen and Gallegos’ definition of FL is widely cited, relying solely on this framework may limit the scope of the review. It would be beneficial to incorporate alternative or complementary definitions and frameworks to provide a more comprehensive analysis of FL interventions.

Thank you for your comment. While there are many FL definition and frameworks, we believe Vidgen and Gallegos’ framework provides the best benchmark in which to assess FL as this time as it is the most widely used definition in the literature and has been assessed as one of the most robust definitions that encompasses the many aspects of FL in both a systematic review by Krause et al in 2018 and a scoping review by Truman et al in 2017. We have however added increased discussion of alternative definitions in the introduction which can be found in highlighted text on pages 1-2, lines 37-74.

The review suggests that all studies showed some improvement in FL outcomes, but it doesn't critically evaluate the magnitude of these improvements. The discussion on the extent of the improvements and the potential limitations in measuring FL outcomes would provide a clearer picture of the effectiveness of these interventions.

The focus on populations at high risk of food insecurity is important, but the review could explore whether these interventions are equally effective across different demographic groups. Are there specific subgroups within these populations that benefit more or less from the interventions? 

Thank you for your comment. These are interesting suggestions however, we believe this level of analysis is outside the scope of this review. The purpose of scoping reviews as outlined by Munn et al in 2018 and Bradbury-Jones et al in 2022, are to map and examine a particular field of research, clarify key concepts or definitions, identify key characteristics of a concept or field of research, identify knowledge gaps and serve as a precursor for a systematic review. We believe that evaluating the size of the effectiveness and comparing the size of the effect across different populations goes beyond fulfilling these aims. However, based on your suggestions, a systematic review that further explores your comments is being planned.

The review highlights the use of FL theoretical frameworks in intervention design but doesn’t sufficiently critique the quality or appropriateness of these frameworks.

Thank you for your comment. While we agree this is an interesting consideration when comparing and contrasting food literacy interventions, this review aimed to map the current evidence surrounding food literacy interventions, describe their characteristics and compare their components to the framework provided by Vidgen and Gallegos. Therefore, providing critique for each FL theoretical framework and analysing its appropriateness for each intervention goes beyond the purpose and aims of this scoping review. The different definitions of FL are now discussed in greater detail in the introduction

Also, the review acknowledges the heterogeneity of study characteristics (e.g., cohort sizes, durations, target populations), but it doesn’t explore the implications of this variability on the generalizability or comparability of results.

Thank you for your comment. We agree and have therefore included additional discussion regarding the heterogeneity of study characteristics. This can be found in the discussion in highlighted text on page 17, lines 387-390.

While the review identifies gaps, such as the minimal inclusion of FL theory in interventions, it should provide more concrete recommendations for future research. What specific aspects of FL theory should be incorporated into future interventions? How can researchers better measure and evaluate FL outcomes?

The review, mainly in conclusions, should also emphasize the need for more rigorous and long-term follow-up studies to confirm these findings. 

Thank you for your comments. We agree and have revised the manuscript to include your recommendations. These revisions can be found in the conclusion in highlighted text on page 17, lines 406-413.

Reviewer 3 Report

Comments and Suggestions for Authors

The paper entitled ‘A Scoping Review of Food Literacy Interventions’ aimed to map the current evidence relating to food literacy interventions, assess their characteristics against the components provided in the definition of food literacy provided by Vidgen and Gallegos and describe their characteristics to identify gaps in the literature These gaps were defined by formulating 4 questions. I think the manuscript may be interesting to a wide audience, is logic and well written.  

Detailed remarks/comments given below can be addressed by authors.

Abstract: I think it would be better to write about 7 instead of 9 studies that had been included according to data given in flowchart of research information in lines 144-149.

Maybe a list of abbreviation would be helpful to readers.

Some abbreviation should appear earlier when the name was mentioned for the first time.

Editorial correction is required.

Position/titles of Authors should be removed from the heading.

Author Response

Thank you for considering our manuscript. We appreciate the feedback provided and have addressed the reviewer's comments below and by making amendments to the manuscript (highlighted in yellow in the document). 

Abstract: I think it would be better to write about 7 instead of 9 studies that had been included according to data given in flowchart of research information in lines 144-149.

Thank you for pointing this out. We agree with this comment, amendments noted  in highlighted text in the Abstract located on page 1, line 19.

Maybe a list of abbreviation would be helpful to readers.

Thank you for your comment. We agree and have therefore included a list of frequently used abbreviations at the start of the manuscript. This can be found in highlighted text on page 1, line 28.

Some abbreviation should appear earlier when the name was mentioned for the first time.

Thank you for pointing this out. We agree and have therefore reviewed all acronyms used in the manuscript and adjusted these to ensure their inclusion complies with the journal guidelines: “Acronyms/Abbreviations/Initialisms should be defined the first time they appear in each of three sections: the abstract; the main text; the first figure or table. When defined for the first time, the acronym/abbreviation/initialism should be added in parentheses after the written-out form.” To highlight these revisions, the first use of all acronyms in each of the three sections has been highlighted in yellow.

 For improved clarity, the acronyms for the terms ‘dietitian’ and ‘social cognitive theory’ have also been added to the Key for Table 2 as these terms are used multiple times throughout the table. These changes can be seen in highlighted text in the Key beneath Table 2 on page 11.

Position/titles of Authors should be removed from the heading.

Thank you for this comment. We agree and therefore, have removed the titles and positions from the authors. This can be found in highlighted text beneath the title of the manuscript on page 1, line 3.

Round 2

Reviewer 2 Report

Comments and Suggestions for Authors

Taking in consideration the modifications and the comments made by the authors, I agree with the publication of the manuscript.